# Size-Dependent Structural, Magnetic and Magnetothermal Properties of Y_3_Fe_5_O_12_ Fine Particles Obtained by SCS

**DOI:** 10.3390/nano12162733

**Published:** 2022-08-09

**Authors:** Tatiana Kiseleva, Rashad Abbas, Kirill Martinson, Aleksei Komlev, Evgenia Lazareva, Pavel Tyapkin, Evgeniy Solodov, Vyacheslav Rusakov, Alexander Pyatakov, Alexander Tishin, Nikolai Perov, Enkhnaran Uyanga, Deleg Sangaa, Vadim Popkov

**Affiliations:** 1Physics Faculty, Moscow M.V. Lomonosov State University, Leninskie Gory, b.1, Str. 2, 119991 Moscow, Russia; 2Saint Petersburg State Institute of Technology, 190013 St. Petersburg, Russia; 3Ioffe Institute, Politechnicheskaya Str., 26, 194021 St. Petersburg, Russia; 4Institute of Solid-State Chemistry and Mechanochemistry RAS, Kutateladze Str., 18, 630090 Novosibirsk, Russia; 5Institute of Physics and Technology, Ulaanbaatar 13330, Mongolia

**Keywords:** garnet, magnetothermal, heat generation ability, cation distribution, X-ray diffraction, Mössbauer spectroscopy, Raman spectroscopy, magnetic measurements

## Abstract

Iron-containing oxides are the most important functional substance class and find a tremendous variety of applications. An attractive modern application is their use in biomedical technologies as components in systems for imaging, drug delivery, magnetically mediated hyperthermia, etc. In this paper, we report the results of the experimental investigation of submicron Y_3_Fe_5_O_12_ garnet particles obtained in different sizes by solution combustion synthesis (SCS) using glycine organic fuel to discuss the interdependence of peculiarities of the crystal and magnetic structure and size’s influence on its functional magnetothermal performance. A complex study including Mössbauer and Raman spectroscopy accompanied by X-ray diffractometry, SEM, and measurements of field and temperature magnetic properties were performed. The influence of the size effects and perfectness of structure on the particle set magnetization was revealed. The ranges of different mechanisms of magnetothermal effect in the AC magnetic field were determined.

## 1. Introduction

Yttrium ferrite Y_3_Fe_5_O_12_ (YIG) with a garnet structure is well known as an attractive material for practical applications in microwave technology, circulators and phase shifters, and spintronics [1,2,3,4,5,6,7]. The possibility for the practical application of ferrite garnets is determined by the presence of a number of effects that manifest themselves in the magnetic, electrical, optical, electromagnetic, and magneto-optical properties. The newly explored applications of this compound are related to several interesting effects arising from the structural distortion caused by the electrical polarizability or hopping charge carriers [8] in the presence of a magnetic field. Another promising effect occurs in fine YIG particles with the redistribution of the spin coupled dipole moments upon the application of an AC magnetic field [9,10]. It results in a functional magnetothermal effect for magnetothermic, ablation [11,12,13,14,15], and photothermal therapy [16] and as a part of organic composite networks [13,17,18].

The YIG crystallographic structure was reviewed in [19,20]. According to known crystallography concepts, YIG belongs to the space group O_h_^10^. There are eight formula units in a cubic YIG unit cell. The trivalent iron atoms occupy the 24d and 16a positions. They are surrounded by oxygen atoms (96 h) with a tetrahedral (for d-site) and octahedral (for a-site) structure. Namely, the structure of YIG has three types of polyhedra: FeO_4_ tetrahedra and FeO_6_ octahedra are joined alternatively by sharing corners and together share edges with YO_6_ dodecahedra to form a three-dimensional framework. Iron nuclei usually occupy tetrahedral and octahedral sites in a ratio of 3:2. Each site has sufficient symmetry to ensure that the iron nuclei EFG tensor is axially symmetric. The later works [21] showed that lattice imperfections and distortion as the consequences of oxygen or metal-ion deficiency might result in the reduction of the unit cell symmetry from cubic to rhombohedral. It was established that the ferrimagnetism of YIG arises from the antiferromagnetic coupling between two magnetic sublattices of Fe^3+^ ions located at tetrahedral and octahedral sites. This strong a–d superexchange interaction constrains the sublattice magnetizations Ma and Md to be antiparallel.

It is known that the magnetic properties of the bulk YIG samples are very sensitive to changes in the local magnetic and structural environment of the lattice sites [22]. The distribution of cations can be the main reason for the change in the saturation magnetization of YIG directly through the ferrimagnetism of the system or indirectly through the exchange path.

The size effects for YIG’s functional properties were reported in [23,24]. The authors propose that the surface layer’s effects on the particle’s magnetic behavior appear like spin-canting behavior.

It was shown that synthesis of the compositionally pure fine YIG particles is a rather difficult task. This process is highly dependent on several chemical factors and the synthesis temperature [24,25,26,27]. Phase separation during synthesis and the formation of even a small number of impurities like iron oxide Fe_2_O_3_ or ternary ferrite YFeO_3_ result in the appearance of the effects of the non-stoichiometry or mixed-valence and metastable states [28,29]. As a result, a noticeable particle size distribution is observed during synthesis, which affects the magnetic properties of the samples. In practical applications, the formation of atomic-scale defects may significantly affect functional performance. Enhancing the influence of defect formation on the magnetic behavior is realized by YIG-directed substitution by Dy, Gd, etc. [30,31] or creating the lattice deformation, which leads to a change in the magnetocrystalline anisotropy.

Small YIG particles synthesized by various methods were recently reviewed [32]: co-precipitation, hydrothermal, sol-gel, pulsed laser deposition, etc. Recent work [33] showed that solution combustion synthesis (SCS) is a standout method for obtaining fine pure YIG particles. At the same time, this makes it possible to obtain a nanostructured product quickly and easily and also makes it possible to regulate the desired ratio of reagents and achieve their homogeneity in the reaction solution. This method is based on the release of thermal energy by an exothermic reaction between an oxidizing agent (nitrate ions) and a reducing agent (fuel) initiated at low temperatures. It has been shown that the choice of an appropriate fuel must be a critical parameter since it is responsible for changing the combustion mechanism and kinetics. Therefore, it makes it possible to control the characteristics of the product. We found that the most homogeneous, nanosized and pure YIG particles may be synthesized using glycine as organic fuel in solution combustion synthesis.

Despite tremendous works related to searching for advanced compounds with better performance of magnetothermal properties for promising application in magnetothermic or ablation of pathogenic cells, there are only a few papers that deal with the study of Y_3_Fe_5_O_12_-based particles [10,12,34]. The high heat generation ability of submicron YIG particles was demonstrated, but the data related to bacterial activity and toxicity of this compound seems controversial. The results obtained on submicron YIG particles [9], synthesized by colloidal chemistry, possessed high heat generation ability but the origin of the observed behavior remained in question because of the absence of a detailed crystalline and magnetic structure study. Experimental investigation of the real small particles with complex magnetic structures and exchange interactions are very effective with methods that demonstrate sensitivity to the local environment of atoms, crystal, electronic, and spin structure and size effects. Mössbauer gamma-resonant spectroscopy and Raman spectroscopy supported by the revelation of crystal structure with X-ray diffraction ensure electronic, magnetic, and crystal self-consistent data may support the understanding of the YIG particle’s functional performance [4,35].

The present research is devoted to the study of the influence of the size, crystalline and magnetic structure of the pure YIG fine particles synthesized by solution combustion synthesis (SCS) using glycine organic fuel followed by temperature treatment on their magnetic properties and functional heat generation performance in altering the magnetic field.

## 2. Materials and Methods

### 2.1. Sample Preparation

The yttrium iron garnet (Y_3_Fe_5_O_12_, YIG) powder sample was synthesized via solution combustion synthesis using yttrium nitrate hexahydrate «Y(NO_3_)_3_·6H_2_O, 99.0%» and iron (III) nitrate nonahydrate «Fe(NO_3_)_3_·9H_2_O, 98.0%» as oxidizers and glycine «C_2_H_5_NO_2_, 99.5%» as a fuel, which was taken in an amount exceeding the stoichiometric one—G/N ratio was equal to 2.0. The starting reagents were dissolved in distilled water with stirring. Then the resulting solution was heated in a heat-resistant beaker on a heating plate to boiling, accompanied by the active evaporation of water and release of large quantities of carbon dioxide and nitrogen gases. After the removal of the main part of the solvent, a gel-like product was formed, which then spontaneously ignited and burned with the formation of a foam-like, highly porous substance. The chemical equation of the reaction can be represented as follows:
9Y(NO_3_)_3_ + 15Fe(NO_3_)_3_ + 40C_2_H_5_NO_2_ = 3Y_3_Fe_5_O_12_ + 80CO_2_ + 100H_2_O + 56N_2_(1)

Fuel (glycine) was taken in an amount exceeding the stoichiometric one—*G/N* ratio was equal to 2.0. Finally, the samples were ground into homogenous powders, which were then calcined in air for 4 h at different temperatures of 800, 900, 1000, 1100, and 1200 °C to obtain pure yttrium iron garnet particles (YIG). Specimens were marked as YIG800, YIG900, YIG1000, YIG1100, and YIG1200.

### 2.2. Methods

The morphology and crystal structure of the prepared samples were studied by scanning electron microscopy (SEM) using a VEGA3 TESCAN scanning electron microscope.

X-ray studies were carried out on a Panalytical Empyrean diffractometer with a copper anode (λ = 1.54 Å, operation modes I = 40 mA, and U = 40 kV). The diffraction patterns of powder samples were recorded in the Bragg–Brentano focusing geometry with a step of 0.026° in the angle range 5–120° using a Ni-filtered diffracted beam and a Pixel3D area detector. The PXRD data were analyzed by HighScore Plus software (PANalytical) and the ICSD structure database. The average sizes of the obtained Y_3_Fe_5_O_12_ crystallites were calculated by broadening the X-ray lines using Scherrer’s formula.

Raman spectra were recorded on a Raman Flex400 (PerkinElmer) spectrometer with fiber-optic probes in reflection geometry (diode laser excitation wavelength 785 nm, linewidth 0.03 nm, and spot size 3 mm). The spectra were recorded in the range from 200 to 1000 cm^−1^ at a laser power of 25 mW. The spectrum acquisition time was 4 s for 20 scans. The baseline was adjusted automatically to the internal reference. PerkinElmer SPECTRUM software was used to process the spectra and determine the positions, intensities, shapes, and widths of the lines.

Mössbauer spectra were recorded using an MS1104Em Mössbauer spectrometer operating in the regime of constant accelerations with a triangular shape of the Doppler velocity of the source relative to the absorber. Spectra were collected in transmission geometry with Co^57^(Rh) applied as the gamma-radiation source. The spectrometer was calibrated with α-Fe foil. The fitting of the spectra was realized in the SpectrRelax Program software [36] within the full Hamiltonian model for crystalline and magnetic structures with a valuable combination of magnetic dipole and electric quadrupole hyperfine interactions [37].

Magnetic properties of the synthesized particles were investigated using the vibrating sample magnetometer (VSM) 7407 Series (LakeShore, Carson, CA, USA) in fields up to 1.6 T. The magnetic moment was normalized to the mass of the samples, and saturation magnetization (Ms) was obtained after fitting the high-field region of the M(H) curve.

Magnetothermal properties were measured on the device according to the procedure [38,39] in an alternating external magnetic field (AMF). Experiments were performed in an experimental setup (AMT&C Group (Moscow, Russia)). The experimental setup included a magnetic module with an inductive coil, connected in circuit with the AC generator and the reconfigurable capacitor system enabling switching of the frequency range. The setup also had a water-cooling system to prevent parasitic heating due to eddy currents and a PC-based data acquisition system comprising the micro-voltmeter Agilent 34410A connected to a thermocouple with one end in the test tube and another one in the reference temperature thermostat. Suspensions of 20 mg of YIG particles with a relative amount of water in 0.5 mL Eppendorf were used. The most commonly quoted measure for the magnetic heating ability of particles—the specific absorption rate SAR = *P*/*m* = C(dT/dt) (M/m)—was calculated from experimental data, where *P* is the heating power, measured in W and generated per unit particles mass *m*, measured in g; where C is the heat capacity of a liquid, dT/dt is the heating rate that can be obtained either by a corrected slope method (the sum of the modulus of the slopes for heating and cooling curves at a fixed temperature), and M/m is the ratio of the mass of water to the particle’s mass.

Due to the high heating rate and approaching the boiling temperature of water, the conventional Box-Lucas method, based on approximation of the heating curve with the dependence *(1 − exp(−t/τ)),* where *τ* is a characteristic time constant of the system, is reliable only for the samples with a slow heating rate, while for nanoparticles with high absorption rate a variation of the corrected slope method was used that takes into account the non-adiabatic measurement conditions [40]. To estimate thermal losses at maximum temperature (in this case, the water boiling point), the magnetic field was switched off, and the cooling curve was measured. The real value of dT/dt at any given temperature can be obtained as a sum of the slopes of the heating and cooling curves [41].

## 3. Results

### 3.1. Scanning Electron Microscopy

Synthesized particles have different morphology and size distribution according to microscopy results. The SEM images shown in Figure 1 indicate that the morphology of the specimen changes from highly porous agglomerates of very fine particles (Figure 1a,b) to larger and well-crystallized ones (Figure 1c–e).

### 3.2. X-ray Diffraction

The results are consistent with the XRD data shown in Figure 2a. The main phase in all samples was cubic triyttrium pentairon (III) oxide (*c*-Y_3_Fe_5_O_12_) with the space group Ia3d. All diffraction peaks of the YIG samples were matched by pure YIG. Only YIG900 contained the reflections of the hematite α-Fe_2_O_3_ in addition to the main phase of garnet ferrite. The appearance of this impurity in the samples, as shown in [33], may be associated with the partial carbonization of combustion products during preparation. The formation of the impurity phase after heat treatment leads to sample inhomogeneity. The calculated lattice constant decreased monotonically from *a* = 12.385 ± 0.002 Å (YIG800) and reached the value *a*= 12.375 ± 0.002 Å (YIG1200) with annealing temperature (Figure 2b). Consequently, the crystalline structure became denser. The particles’ size distributions were governed by lognormal dependence (Figure 2c). The average crystallite sizes of the particles via the temperature of the synthesis are shown in Figure 2c, (insert). The values of the average sizes approximated from diffraction linewidth by lognormal distribution were larger, and the maxima for each particle’s size distribution marked as D_max_ were found to be 35, 50, 83, 137, and 187 nm, corresponding to the YIG800-YIG1200 samples (See Table 1). A narrower particle distribution was found for YIG800.

### 3.3. Raman Spectroscopy

Figure 3 shows the Raman spectra with the band assignment based on the known predicted positions of active vibration for the garnet [14]. Raman active phonons belonging to the 3A_1g_ + 8E_g_ + 14T_2g_ active modes for the YIG structure were clearly resolved for all samples. The results demonstrated the high purity of the synthesized samples since the specimen was measured using a fiber probe with a diameter of 2 mm and the repetition of spectra from the different areas of the powder. The presented Raman spectra show that the YIG vibrational modes, most of which are in the region of lower Raman shifts (300 cm^−1^), are usually associated with Y^3+^ and [FeO_4_]^5−^ ionic particles and labeled as their translations.

Peaks (E_g_) and (A_1g_) represented two internal modes of the [FeO_4_]^5−^. The frequency shift of the characteristic phonon modes to the high value and corresponding reflections broadening with annealing temperature clearly indicated the effect of the particle size decrease. Particle size effects were delivered from the characteristic vibration mode shifts to the higher frequencies and linewidth broadening.

### 3.4. Mossbauer Spectroscopy

Mössbauer spectra measured at room temperature are shown in Figure 4. Their profile is typical for YIG but has peculiarities characteristic of crystalline and magnetic structure and from the effects of small sizes. The broadening of the spectral lines and anisotropy of the line profile correspond to the set of the nonequivalent iron position in garnet sublattices. The main part of all spectra was presented by several subspectra corresponding to the nonequivalent iron sites in the yttrium iron garnet structure and has been fitted using a model of full Hamiltonian due to considerable combined hyperfine interactions with a high value of quadrupole interaction relative to the magnetic dipole interaction.

Garnet phase model fitting was carried out because its ferrimagnetism is generally explained by the existence of two nonequivalent magnetic sublattices composed of iron atoms in (16) octahedral and (24) tetrahedral sites coupled antiferromagnetically. The Mössbauer spectrum of single crystal YIG usually consists of subspectra corresponding to several octahedral (*a*) (green colored) and tetrahedral (*d*) (blue colored) structural sites of Fe^3+^ in oxygen surroundings [22], which amounts and parameters are determined by site population, angles θ between electric field gradient tensor principal axis, and the easy magnetization axis. We assume that synthesis resulted in the formation of the garnet phase with several crystal and magnetic structure distortions explaining the observed number of subspectra components, their parameters, and mutual relation. The distortions [23] rotate the principal axis of the electric field gradient tensor relative to the [111] direction resulting in four different θ angles. The origin of distortion may be the consequence of oxygen or metal-ion deficiency resulting in changes in the local structural configuration [21]. Our model for spectra fitting consists of three subspectra for octahedral Fe^3+^ surrounding and four subspectra for tetrahedral Fe^3+^ surrounding for the volume of the particles (V) and two additional subspectra corresponding to nonequivalent iron positions on the particles’ surface (S). The Mössbauer spectra with resolved subspectra are shown in Figure 5. The hyperfine parameters of subspectra derived from model fitting are collected in Figure 5 and Figure 6, dependent on Dmax particle sizes. The values of hyperfine magnetic fields for octa- Fe^3+^ positions H_hf_(octa-V) and tetra- Fe^3+^ positions H_hf_(tetra-V) for the YIG800-YiG900 particles’ volume were smaller than the characteristic value for the bulk YIG (Figure 5a). Specifically, the corresponding hyperfine fields associated with Fe^3+^ ions on the particle’s surface H_hf_(octa-S) and H_hf_(tetra-S) were considerably reduced due to uncompensated bonds and structural defects. The reduction of the isomer shifts’ (δ) average values for *a*- and *d*- sites are explained by densification and the particle crystal structure becoming more perfect (Figure 5b) supported XRD structural data on decreased lattice parameter.

Moreover, the increase in linewidths was observed for all components, resulting from the presence of local inhomogeneities due to the statistical distribution of nonmagnetic irons around the Fe^3+^ occupied sites. For all synthesized specimens, the wider distribution of H*_hf_* was derived for octahedral sites rather than tetrahedral. The distribution of Fe^3+^ in sublattices was determined using Mössbauer fitting results according to the relation *n*_d_/*n*_a_ = x = (*S*_d_/*S*_a_)(*f*_a_/*f*_d_), where *f*_a_/*f*_d_ = 0.94 is the probability of the recoilless nuclear resonance absorption of gamma radiation for *a*- and *d*- sites at room temperature [22]. The deviation from the theoretical value for perfect structure x = 1.5 was observed, and reducing the value with annealing temperature was determined (Figure 6a). The surface-to-volume amount of iron ion states in the particles (Figure 6b) was determined to decrease with particle size, increasing practically twice from YIG800 to YIG1200.

### 3.5. Magnetic Measurements

The magnetic behavior of the YIG nanoparticles investigated at room temperature is shown in Figure 7. All samples demonstrated ferromagnetic hysteresis. The coercive force is nonzero for all hysteresis loops, and the saturation field is greater than 1000 Oe.

The dependence of coercive force *(H*_c_), saturation magnetization (*I*_s_), and residual magnetization (*I*_r_) values vs. the particle sizes are shown in Figure 8. The nonmonotonic dependence of the coercive force of samples on their average size was observed (Figure 8a). The maximum coercive force *H*_c_ was observed for YIG1000, which can be explained by a change in the magnetic state of the particles. An increase in the average particle size of the sample leads to a decrease in the magnetic anisotropy of the surface. Consequently, the particle passes from a single-domain to a multi-domain state, which is characterized by a lower coercive force. The decrease in the coercive force for samples YIG800 and YIG900 may be associated with the presence of superparamagnetic particle fractions that have zero coercive force. The critical diameter D_cr_(sp) for YIG nanoparticles at which the transition to the superparamagnetic state occurs corresponds to 35 nm [42]. Therefore, only YIG800 demonstrates visible superparamagnetic behavior. Another possible reason, which may explain such coercivity behavior, is the formation of a slightly oxygen deficient structure due to the synthesis course. The slight deviation of the isomer shift (marked by a red arrow in Figure 5b) indicates a change in the electron density on the octahedral site. We assume that particles with a diameter close to Dcrit~100 nm are predominantly in the single-domain state. This result is in good agreement with the literature data [42]. It should be noted that the samples are not monodisperse. Therefore, each sample contains particles in different micromagnetic states. Consequently, we can only estimate that most of the particles in the sample are in a different micromagnetic state.

The results of the residual magnetization dependence on the particle size (Figure 8c) correlate with the behavior of the coercive force. Figure 8b shows the dependence of the saturation magnetization on the average particle size. Saturation magnetization values for all samples are smaller than the saturation magnetization for the bulk YIG (27.40 emu/g—3.62 Bohr magnetons) at 292 K. This fact can be explained by the presence of a canting of the magnetic moment on the surface of the particles. We assume that an increase in the saturation magnetization for samples YIG800, YIG900, and YIG1000 is associated with a decrease in the superparamagnetic particle’s fraction. Another possible reason for the increase in the saturation magnetization is a decrease in the fraction of surface atoms with a canted magnetic sublattice. The nonmonotonic behavior in the D max size range from 80 to 200 nm should be noted. The local minimum of the saturation magnetization at an average diameter of 138 nm (and sharp decrease at low temperatures) may be explained by the presence of cation distribution changes between sublattices, as noted earlier in the discussion of Figure 5b.

An increase in the saturation magnetization with an increase in the average particle size can be explained by a decrease in the fraction of near-surface atoms compared to the fraction of atoms contained in the bulk of the particle. This supports a finding derived from Mössbauer spectra analysis with decreasing the canted spin states at the particle’s surface (Figure 6a).

The temperature dependences of the magnetization in a small field (50 Oe) are shown in Figure 9a. The value of the external magnetic field is chosen to be less than the value of the coercive force in order to be able to study the relaxation processes of magnetization. All samples, except YIG800, show the same course of heating (ZFC—zero field cooled) and cooling (FC—field cooled) curves. The differences in the ZFC-FC curves for sample YIG800 can be explained by the presence of a significant fraction of superparamagnetic particles in this sample. From the divergence of these curves, we can estimate that the blocking temperature corresponding to the transition from the superparamagnetic state to the ferromagnetic state for the YIG800 is 385 K. All other samples do not demonstrate visible superparamagnetic behavior. It is also obvious that the Curie temperature Tc for all samples is above 450 K.

We measured the Tc values from the temperature dependence of susceptibility (Figure 9b) based on the Hopkinson effect. Therefore, the magnetization of a sample is determined by its initial susceptibility. The samples have different initial susceptibilities and magnetization in a field of 50 Oe since these parameters are determined by the values of the Zeeman energy, the exchange energy, and the anisotropy energy, which vary for particles with different diameters. The highest magnetization in Figure 9a is exhibited by sample YIG900, which apparently has the highest content of single-domain particles. An increase in the particle size leads to an increase in the stray fields. Therefore, samples YIG1100—YIG1200 have a lower magnetization in Figure 8c.

The effect of the bevel (spin canting) of the magnetic sublattice on the initial susceptibility should also be taken into account. An increase in particle size suppresses this effect, which is accompanied by an increase in magnetization for samples YIG1100 and YIG1200 compared to the magnetization of sample YIG1000.

### 3.6. Heat Generation in AC Magnetic Field

Heating and cooling curves for the YIG particles at various amplitudes of the applied magnetic field are shown in Figure 10. As seen in Figure 10, the heating of the particles was dependent both on the applied field parameters and the particle sizes. Smaller particles, YIG800, YIG900, and YIG1000, showed higher and faster heat generation under all applied field amplitudes and frequencies (the highest rise in the temperature up to 80 °C within several min), while YIG1100 was heated slowly.

The specific absorption rate (SAR) for a specimen calculated from the measurements is presented in Figure 11a. The behavior of the heating rate for samples indicates that energy losses for samples are proportional to the different indicators of the exponent via the magnetic field value. The reason for the observed decrease in heat generation ability with the increase in the particle size ascribed to the coercivity Hc for the hysteresis loss is proportional to the reciprocal of the particle diameter (1/Dmax) and transition from one magnetic state to another. SAR behavior in synthesized particles may be divided into two ranges according to known mechanisms of particles heating in the AC field. 

For samples YIG800, YIG900, and YIG1000 due to the larger width of hysteresis curve (Figure 7) and in the field larger that coercive field the prevailing mechanism of heating is the hysteresis loss while that correlates with 4th or even 5th power dependence of SAR(H) for these particles (Figure 11). For particles YIG1100 and YIG1200 the heating is driven by more complex mechanism including hysteresis losses of multidomain state-particles as well as relaxation behavior of smaller particles (with quadratic dependence of SAR(H) and their magnetic interactions. Such behavior was discussed in [40], where the authors showed that interacting magnetic particles might exhibit a complex mechanism of local heat transfer since the “particles do work on each other as well as having work done on them by an applied external field, and this work is highly path dependent”. Interacting particles produce heat not only when their magnetization irreversibly switches between two energy wells but also when experiencing intra-well magnetization dynamics.

Namely, when one particle switches, another necessarily has a changed dipolar field, and its magnetization, therefore, has done work which can be converted to heat. This complexity does not allow a simple interpretation of the produced heat in terms of the local hysteresis loops.

## 4. Conclusions

Single phase YIG particles of different sizes were synthesized by solution combustion using glycine fuel followed by heat treatment at different temperatures. The result of the synthesis was pure particles of Y_3_Fe_5_O_12_ with a garnet structure with a particle size range between 15 to 450 nm and average sizes from 35 to 185 nm. The synthesized particles showed different magnetic behavior and demonstrated ranges of size-dependent heating in an applied AC magnetic field with parameters of f = 300 kHz and H~50–70 Oe, corresponding to different domain state transitions and the wideness of polydispersity. The polydispersity of the synthesized particles results in the distribution of different micromagnetic states with very strong exchange interactions. It may also be advantageous to tailor the influence of the magnetic field frequency on heating ability performance for the YIG particles. Such studies are currently underway.

## Figures and Tables

**Figure 1 nanomaterials-12-02733-f001:**
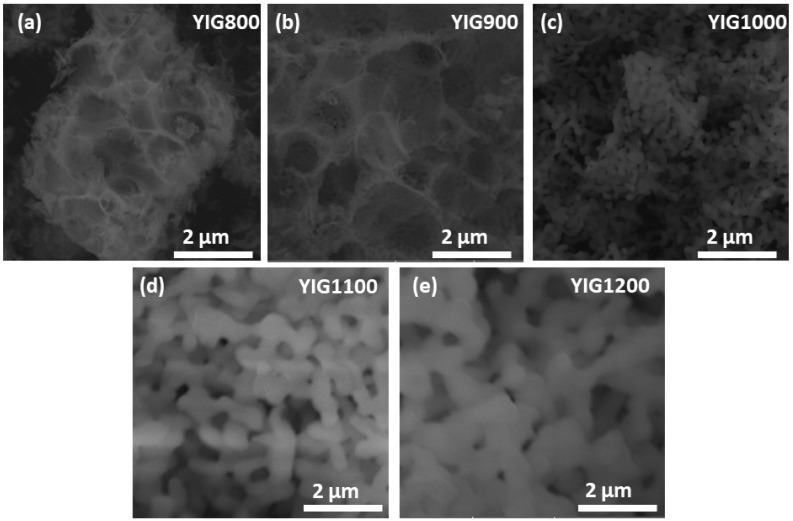
SEM images of YIG particles synthesized at 800 (**a**), 900 (**b**), 1000 (**c**), 1100 (**d**), and 1200 (**e**).

**Figure 2 nanomaterials-12-02733-f002:**
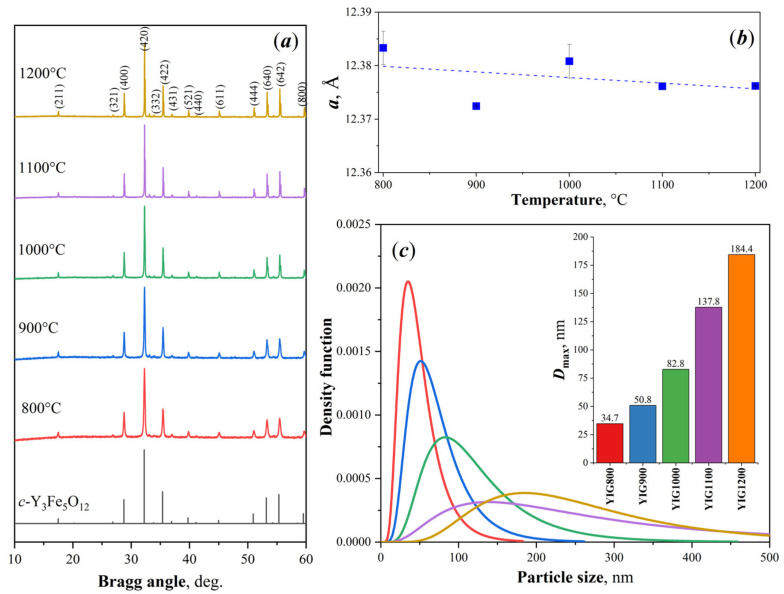
XRD patterns of the obtained heat-treated particles (**a**), lattice constant (**b**), particle’s size distributions from SEM (**c**), and D_max_-crystallite sizes (insert).

**Figure 3 nanomaterials-12-02733-f003:**
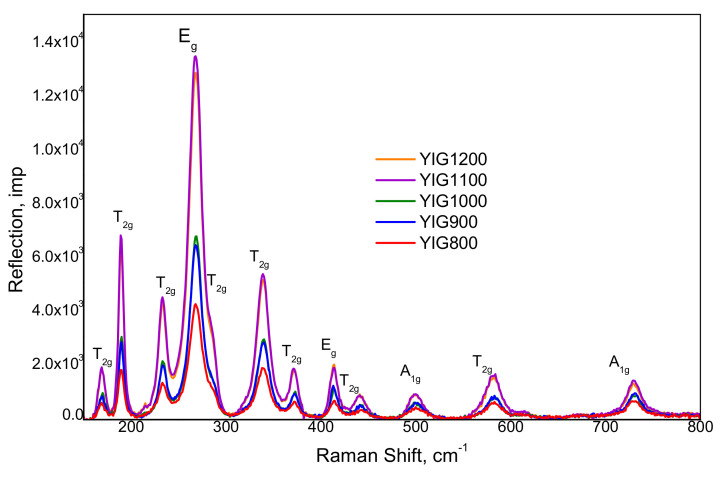
Raman spectra of synthesized YIG particles.

**Figure 4 nanomaterials-12-02733-f004:**
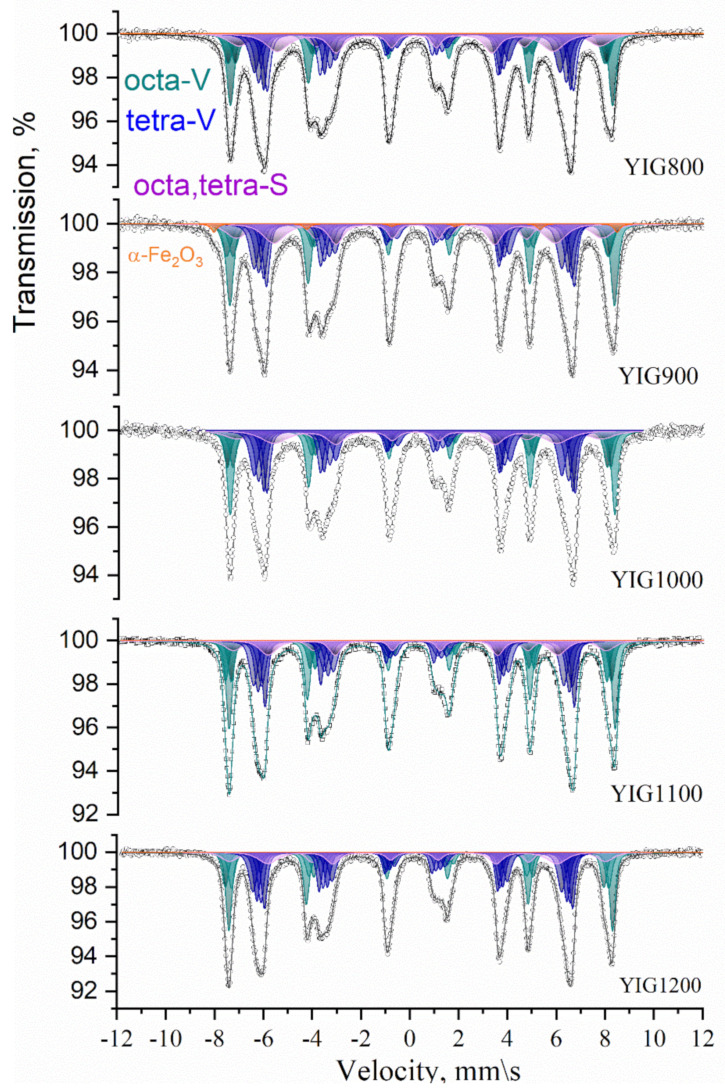
Mössbauer spectra of YIG particles measured at 300 K (octa-green and tetra-blue).

**Figure 5 nanomaterials-12-02733-f005:**
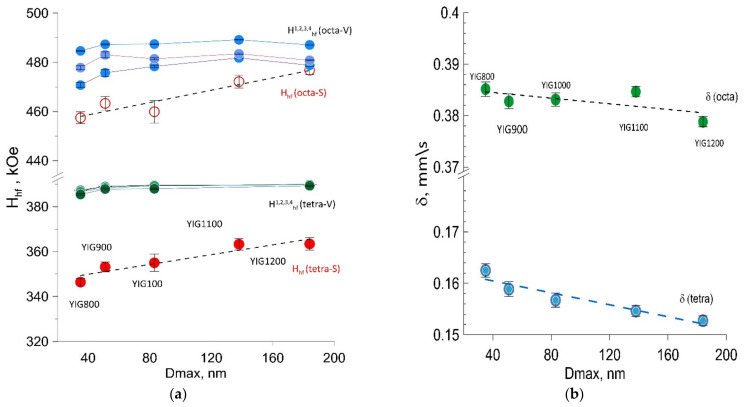
Hyperfine magnetic fields (H_hf_) (**a**) and isomer shift (δ) (**b**) dependences on particle size.

**Figure 6 nanomaterials-12-02733-f006:**
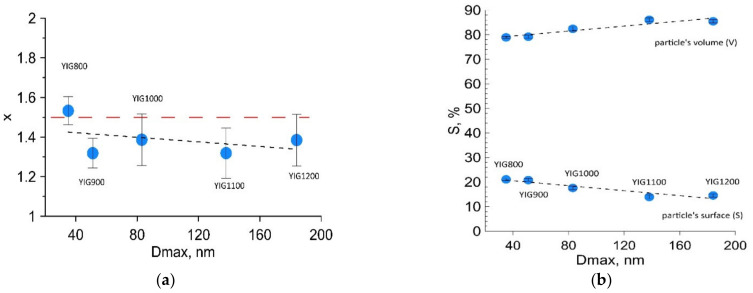
The degree of inversion (**a**) and the amount of surface and bulk fraction of iron in Mössbauer spectrum (**b**) dependences on particle size.

**Figure 7 nanomaterials-12-02733-f007:**
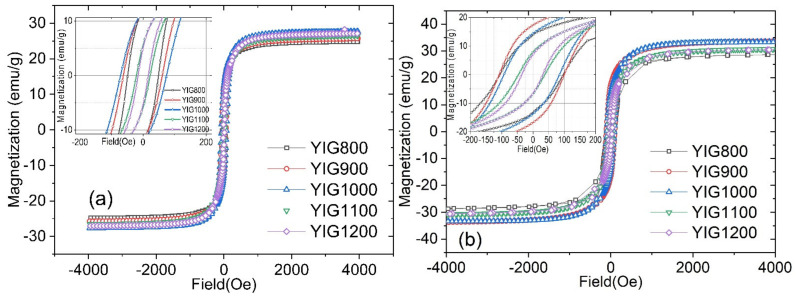
Hysteresis loops at 300 K (**a**) and 90 K (**b**) for synthesized samples.

**Figure 8 nanomaterials-12-02733-f008:**
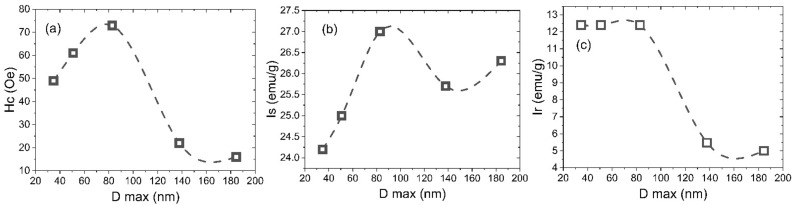
Particle size dependencies on coercivity (**a**), saturation magnetization (**b**), and remnant magnetization (**c**) at room temperature.

**Figure 9 nanomaterials-12-02733-f009:**
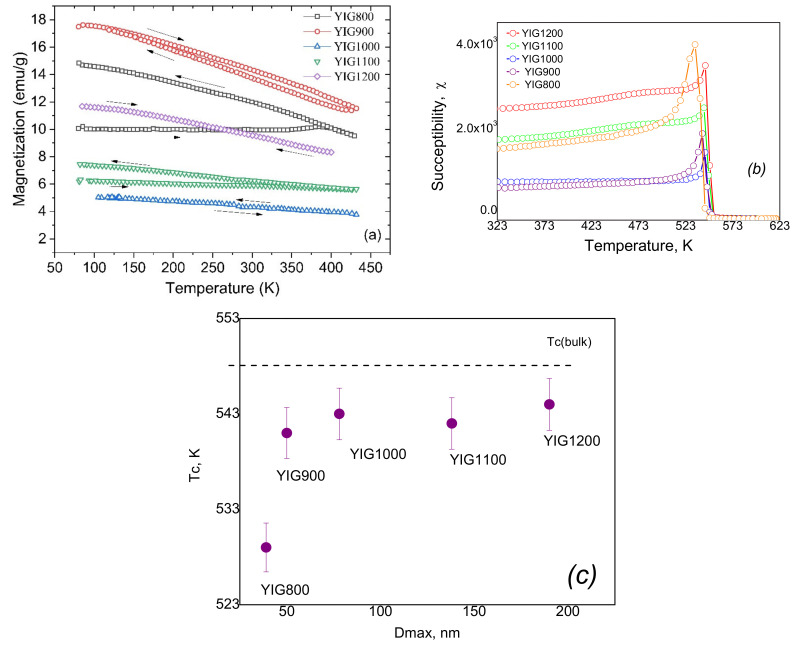
The temperature dependencies of the magnetization in a small field (50 Oe) (**a**), temperature dependencies of susceptibility (**b**), and Curie temperature (**c**).

**Figure 10 nanomaterials-12-02733-f010:**
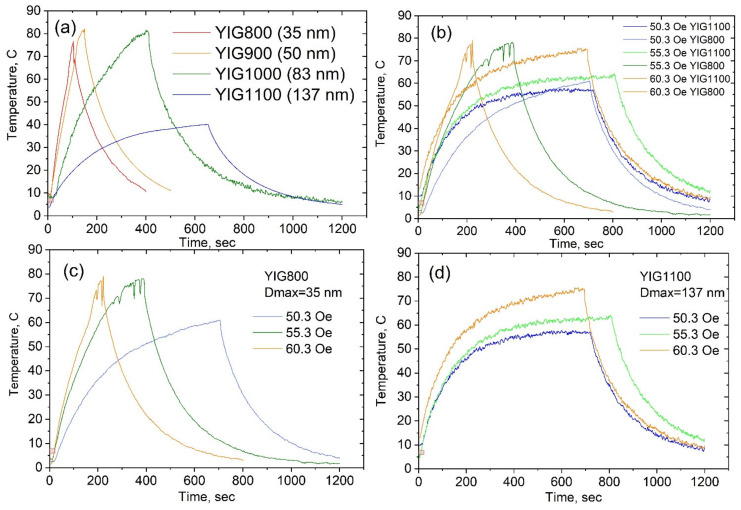
The particle temperature change at AC magnetic field with amplitude H = 70.4 Oe (**a**), the particle temperature change for two sizes (35 and 137 nm) at different H (**b**), the particle temperature change for YIG800 at different H (**c**), and the particle temperature change for YIG1100 at different H (**d**). Increasing—magnetic field switched on; decreasing—magnetic field switched off.

**Figure 11 nanomaterials-12-02733-f011:**
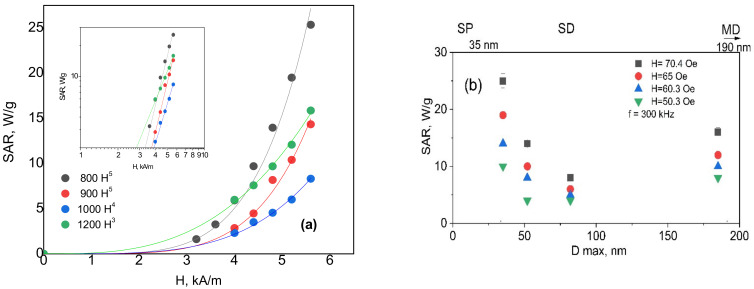
The dependencies of estimated SAR on magnetic field amplitude at a frequency of 300 kHz (the insert—the logarithmic scale) (**a**) and on particle sizes at different magnetic field amplitudes (**b**). SP—superparamagnetic state, SD—single domain state, and MD—multi-domain state.

**Table 1 nanomaterials-12-02733-t001:** Temperature of sample synthesis and particle properties.

Samples	Synthesis Temperature, T, °C	Particles Size, Dmax, nm	Saturation Magnetization, Is, emu/g	Curie Temperature, Tc, °C	SAR, W/g (H = 70 Oe f = 300 kHz)
YIG800	800	35	24.2	256	25
YIG900	900	51	25.0	267	14
YIG1000	1000	83	27.0	270	8
YIG1100	1100	137	25.7	269	16
YIG1200	1200	184	26.3	272	25

## Data Availability

Not applicable.

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
