# Peer review of "Size-Dependent Structural, Magnetic and Magnetothermal Properties of Y3Fe5O12 Fine Particles Obtained by SCS"

_nanomaterials, 2022, doi:10.3390/nano12162733_

Round 1

Reviewer 1 Report

In the present work, authors reported the synthesis of submicron Y3Fe5O12 garnet particles by using solution combustion synthesis method, and then investigated their structural and magnetothermal performance. The results indicated that the size effects and perfectness of structure influenced the particle set magnetization. Overall, the structures were characterized well and the magnetic results were well discussed. However, some issues should be addressed.

1, The abstract can be polished and improved. The novelty problem statement described by the authors should be emphasized to attract general readers by providing more insights on the experimental observations. Also, the authors should elaborate the general applicability of the current work.

2, Authors may rearrange/polish the text and elaborated " Sample preparation" section the way so anybody can repeat the procedures, like a recipe. If there is process flow diagram can be added in fig. 1, it would be helpful to non-specialist readers (optional).

3, The images of Figure 1 were obscure. I suggest authors replace them by high-resolution ones so that readers can find more useful information.

4, In XRD section, authors should index all the peaks in these plots. In addition, the Raman peaks should also be labelled in these plots so that readers would figure out the difference and advance of samples.

5, Controllable and rational processing is a determinant in structure and morphology of nanoparticles. How to improve the controllability and designability of nanoparticle preparation in this work? The authors should also pay attention to this challenge, and some pioneering and original researches about controllable assembly of nanoparticles are suggested: ACS Applied Materials & Interfaces, 2017, 9, 16404; Giant, 2021, 8, 100076.

6, Authors fabricated Y3Fe5O12 garnet particles by using solution combustion synthesis method. The control of various composition of loading Y3Fe5O12 garnet particles is very important. I am wondering how its composition is controlled. I suggest you make a component analysis, such as spectroscopic analysis, since a little difference in composition dramatically influence the magnetic property.

7, Magnetic properties were well discussed. However, I noticed that the nonmonotonic dependence of the coercive force of samples on their average size was seen, and YIG1200, YIG1100 AND YIG800 showed small coercive force, implying an obvious superparamagnetic behavior. However, only YIG800 was confirmed to by superparamagnetic behavior. So why? Please explain this.

Author Response

Dear reviewer!

We appreciate the time and effort that you dedicated to providing feedback on our paper entitled “Size-Dependent Structural, Magnetic and Magnetothermal Properties of Y3Fe5O12 Fine Particles Obtained by SCS (Nanomaterials-1846502)”. We have made corresponding revisions according to your advice. Our response to your remarks, questions and suggestions are listed as follows in the pdf file. We believe that made according your advice corrections will make the manuscript acceptable for publication and it will attract the interest of readers. The text of the manuscript was checked by native English and all found misprints were corrected (the made corrections are marked with a yellow color).

Reviewer 2 Report

The authors need to explain more of the details of their experimental methods. Regarding diffraction, what was the size of the divergence slit, what was the instrumental correction to the width, and what were the details of the refinements such as site occupancy? 

The small Raman peak near 290 cm-1 may be due to Fe2O3.  That may also be the reason for the small unidentified peak near 34 degrees in the diffraction pattern.  The authors need to address that. 

The authors need to discuss what method they used to determine TC

It is odd that the YIG1100  shows such a drop in the saturation magnetization with lowering temperature.  That needs to be discussed or checked.

In Fig. 9, temperature units switch between celsius and kelvin.  The lack of consistency can seem misleading.

The authors should consider a log-log plot for Fig. 11 a.  That might help make the power law dependence more meaningful.

If the authors address all these issues, then the paper should be published.

Author Response

Dear reviewer!

We appreciate the time and effort that you dedicated to providing feedback on our paper entitled “Size-Dependent Structural, Magnetic and Magnetothermal Properties of Y3Fe5O12 Fine Particles Obtained by SCS (Nanomaterials-1846502)”. We have made corresponding revisions according to your advice. Our response to your remarks, questions and suggestions are listed as follows. The text of the manuscript was checked by native English and all found misprints were corrected (the made corrections are marked with a yellow color).

Reviewer 3 Report

In this work, the authors studied in detail the size-dependent structural, magnetic and magnetothermal properties of Y3Fe5O12 Fine Particles Obtained by SCS, by performing Mössbauer and Raman spectroscopy accompanied by X-ray diffractometry, SEM, and measurements of field and temperature magnetic properties. They revealed the influence of the size effects and perfectness of structure on the particle set magnetization, and determined the ranges of different mechanisms of magnetothermal effect in the AC magnetic field. This work is interesting and practically important, and it can be considered for publication after a minor revision:

1. There are some long sentences which are hard to understand, for example, the third sentence in the abstract part. Spell-check and stylistic revision of the paper are necessary. There are also some misspellings throughout the text, for example, “Fig1” on page 4, “Fig5b” on page 9, etc.

2. There have been some relevant works such as Results in Physics 14, 102397 (2019) and Nanomaterials 11(1), 63 (2021). These works should be properly cited to further improve the quality of this work.

Author Response

Dear reviewer 3

We appreciate the time and effort that you dedicated to providing feedback on our paper entitled “Size-Dependent Structural, Magnetic and Magnetothermal Properties of Y3Fe5O12 Fine Particles Obtained by SCS (Nanomaterials-1846502)”. We thank you for your high estimation of our results. We have made revisions according to your advice. The text of the manuscript was checked by native English and all found misprints were corrected (the made corrections are marked in the text of the manuscript with a yellow color). As about the references which you recommended, in our opinion they are a little bit far from the paper topic, sorry!

Round 2

Reviewer 2 Report

The authors have done a careful job of addressing concerns that were raised. It would be best for them to include the information regarding the divergence slit when discussing x-ray diffraction.